# Implementing Encrypted Private Data Sharing between Personal Data Vaults

Anushka Vidanage[1,*,†], Jess Moore[1,†], Dawei Chen[1,†], Tony Chen[1,†], Sergio José Rodríguez-Méndez[1] and Graham Williams[1,†]

[1]*The Australian National University (ANU), Canberra, ACT, Australia*

### Abstract
Adoption of the open Solid specification is emerging as a paradigm shift providing a people-centred model for data storage and ownership, where individuals control access to their data, enabling data sovereignty to be returned to people. By applying a Zero Trust approach to security of data, the Australian National University (ANU) Software Innovation Institute (SII) has designed a Trust No One (TNO) environment for storing data on personal online data stores using encryption and public key cryptography. This TNO environment has been implemented by the ANU SII as a part of two collaborative projects in the health sector that required secure data storage and linked data infrastructure, provided on a basis of data sovereignty and privacy using personal data vaults.

### Keywords
Solid, personal data vaults, encryption, data privacy, consent

## 1. Introduction

Centralised databases remain the core of many applications in industry and research despite being highly vulnerable to security breaches. Security improved with the introduction of access control and other security regulation introduced in the US in 1996 with the Health Insurance Portability and Accountability Act [1]. In the paradigm shift from a centralised web to a highly decentralised web using Solid[2] where individuals own and control personal online datastores (Pods), it is preferable that a Trust No One Environment[3] is adopted as the security model. This paper describes how ANU Software Innovation Institute (ANU SII) has implemented a Trust No One (TNO) environment in digital apps built on Pods with the W3C Solid standard for web decentralisation.

## 2. System design

In a data infrastructure built on personal data vaults, there are multiple layers of security protecting a vault owner's data. Authentication with a webId using OpenID Connect (OIDC) forms the first layer of protection of a person's Pod. Fine-grained access control to files or resources forms the second layer of protection, defining whether a user with a particular webId can access any data file. Encryption of the file contents forms the third layer of protection, and enables the creation of a TNO digital environment on Pods. The ANU SII apps built on Solid architecture implement a TNO environment by ensuring:

*The 4th Privacy & Personal Data Management Session, co-located with the 4th Solid Symposium, April 30 - May 01 2026, London, UK*

*Corresponding author.

†These authors contributed equally.

✉ anushka.vidanage@anu.edu.au (A. Vidanage); Jessica.Moore@anu.edu.au (J. Moore); dawei.chen@anu.edu.au (D. Chen); Yang.Chen2@anu.edu.au (T. Chen); Sergio.RodriguezMendez@anu.edu.au (S. J. Rodríguez-Méndez); graham.williams@anu.edu.au (G. Williams)

🌐 https://comp.anu.edu.au/people/anushka-vidanage/ (A. Vidanage); https://comp.anu.edu.au/people/jessica-moore/ (J. Moore); https://comp.anu.edu.au/people/dawei-chen/ (D. Chen); https://comp.anu.edu.au/people/sergio-rodriguez-mendez/ (S. J. Rodríguez-Méndez); https://comp.anu.edu.au/people/graham-williams/ (G. Williams)

🆔 0000-0002-5386-5871 (A. Vidanage); 0000-0003-1230-6608 (J. Moore); 0000-0001-6063-2622 (D. Chen); 0000-0001-7203-8399 (S. J. Rodríguez-Méndez); 0000-0001-7041-4127 (G. Williams)

- Data stored in Pods is encrypted.
- Data is only ever decrypted in app locally on device.
- No master key is stored in the Pods.
- Access to data is protected by owners and controllers using Access Control Lists or Access Control Policies.

Our properties for a TNO environment is informed by the concept of the Zero Trust Security Model. This is a privacy and security architecture where a system does not automatically trust anyone or anything inside or outside its perimeters and instead must verify everything constantly. Microsoft defined three Zero Trust Principles [4] which are:

- Verify explicitly.
- Use least-privilege access.
- Assume breach.

Figure 1 describes the information architecture for a TNO environment in Pods. Data in a Pod is encrypted using AES encryption [5] with random session keys and data is decrypted only on device locally when using an application. Data sharing is done through public/private key pairs [6], where the sender uses the receiver's public key to encrypt and share the random session key of a document, i.e. only the receiver can decrypt the document. This public key infrastructure for data sharing of with files encrypted by default is implemented in the **solidpod** free open source software package [7] developed by ANU SII (and available with MIT license) for interacting with Pods with the Flutter web framework.

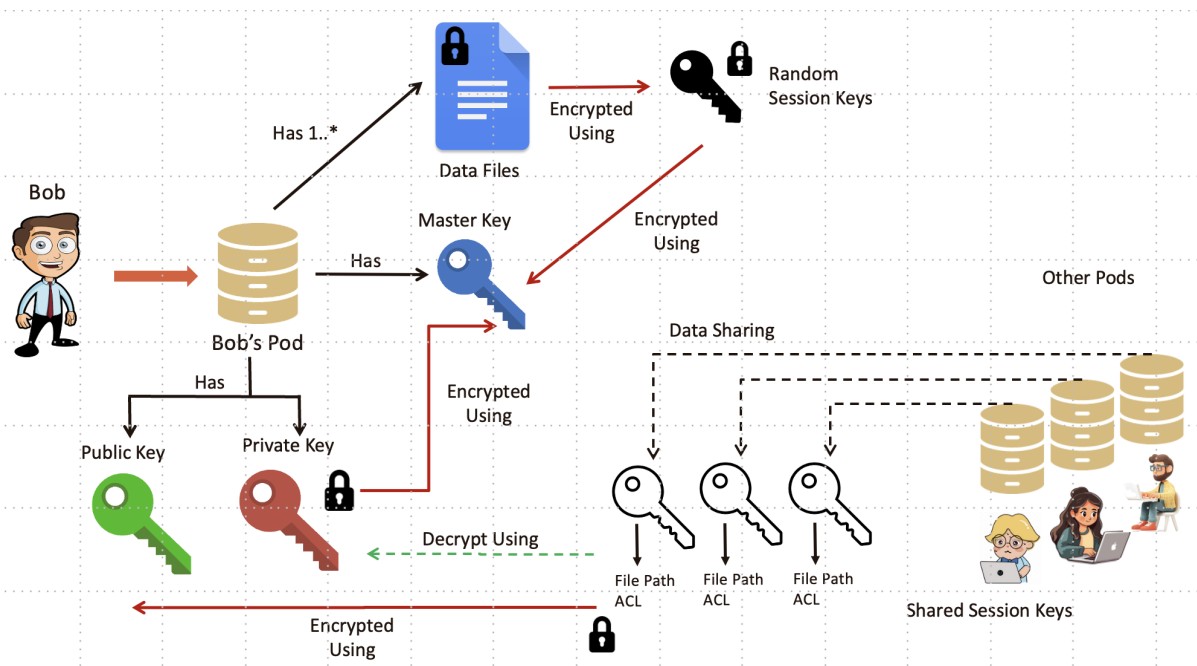

**Figure 1:** Public key architecture of a Trust No One environment to share access to encrypted data files in Pods between different users.

## 3. Data access permissions and consent

Managing data sharing consent is particularly problematic in the context of digital apps, where citizens typically lack specific knowledge, access or control of the volume of personal data held about their data, which is stored in centralised datastores usually owned by the digital app owner. This is an identified ethical issue in medical research clinical trials, where individuals do not fully perceive and understand what type of data about them is being stored, how it is being used, or what data they can

ask to access [8]. The Australian Privacy Principles present four elements of consent in the sharing of personal information: that it be 1) voluntary, 2) informed (before consenting), 3) specific and current, and 4) that the individual has capacity to give consent. [9] Arguably these characteristics of consent are not met when a person's user experience with most digital apps. Typically they are considered to have consented to sharing of their data when they open the app for the first time, and agree to the terms of use. In that circumstance, consent is not freely given. A person must consent to the app terms to use any service offered by the app. The consent is not specific or informed, because users of the app do not know all the data that the app has been collected about them, nor the applications their data is used for, and whether their data is shared to third parties. Furthermore, most users are not capable of consenting as the terms are not readily interpretable for most people.

In the last five years, dynamic consent management systems have been used with centralised data-stores in precision medicine and genomic research projects [10], for example the CTRL web-based application [11], enabling participants to manage their consent for secondary data uses, however whilst

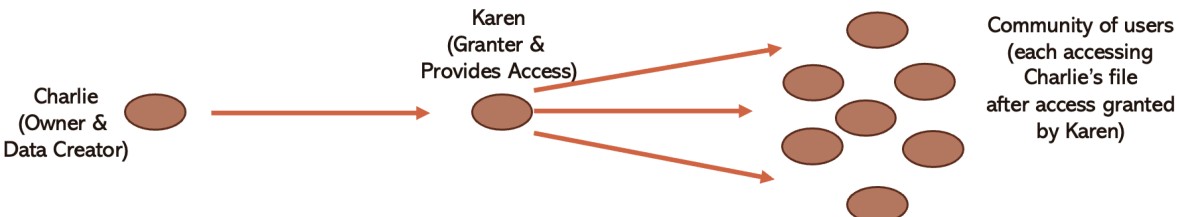

(a) Displays the scenario where Charlie, a creator/owner of a file, grants access to that file to Karen with 'read' and 'control' access. And Karen subsequently grants 'read' to a group of people. For example, Charlie could be a writer, who has a publisher/distributor Karen, who shares access to Charlie's latest article to paid subscribers.

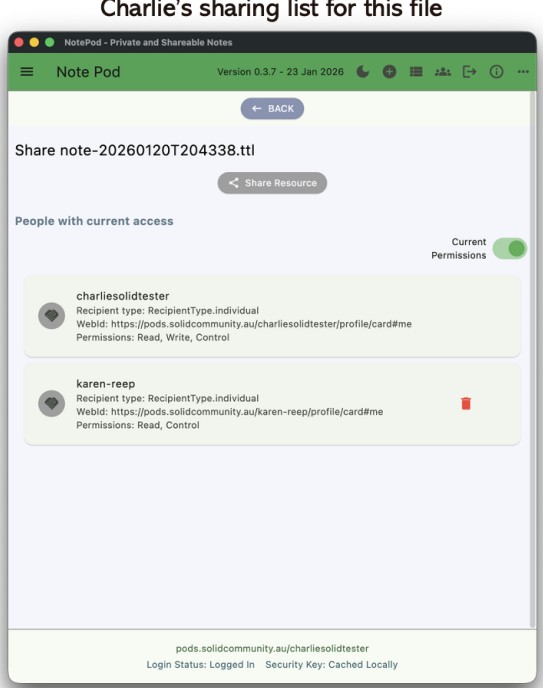
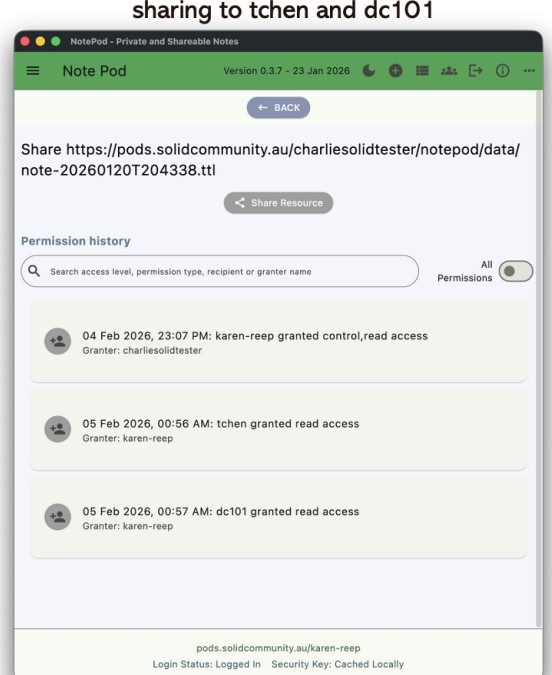

(b) The left view shows Charlie's current access list after they have shared control and read access to their file to Karen. The right screenshot shows Karen's permission history, after they have granter access to two other people in this case tchen and dc101. Charlie's permission history is also updated in this process, and would also show access to users tchen and dc101, which they can revoke if desired.

**Figure 2:** A use case for granting permission to a file in a Pod.

these systems provide participants with broad control over their data, they do not provide participants with access to their data. In instances where data access has been provided to participants, it is usually at the end of a trial. After clinical trials, participants typically lose access and have no further access to their collected data. In all these instances, participants have not had sovereignty of their data and must trust the researchers and/or trial sponsors to make appropriate decisions about the use and reuse of their sensitive data. Such data collection also inherits all the well known privacy and security weaknesses of centralised databases, and is particularly vulnerable due to the sensitive nature of the collected data.

Using Solid Pods, we have provided a TNO environment where citizens have dynamic, granular control of who has access to their data using access control lists with different access levels read, write, append, and control. To provide people with demonstrated control and an easy to understand view of their controls, we have implemented a sharing page allowing a user to switch views between the current access permissions and history of permission granting and revoking records. This provides the file owner and they delegate to have 'control' an auditable record of permissions, that they can query in different ways, such as people with access in a particular date time range or with particular access permissions, or searching for all granting and revoking records to a particular recipient person.

Figure 2 illustrates a scenario where the owner of a file has shared their file to another person who releases that file privately to selected other people. Each user only shares the part of the permission history written to their Pod. The owner of the file sees the full permission history and can grant, change or revoke access permissions to any other recipient. People who've been authorised 'control' access on the file, can also grant, change or revoke access permissions on that file to others. We implement file access sharing and view or search of permission history as part of the **solidui** free open source software package [12] developed by ANU SII (and available with MIT license). The solidui package includes data resource sharing user interfaces for people who own or have delegated control of files in Pods with a transparent interface to manage data access, to grant access to new recipients, change access as they see fit, or revoke access if they no longer authorise permissions to that recipient.

In the ANU SII implementation of a TNO Pods environment, the owner or other authorised controllers of the file, are providing dynamic consent in data sharing transactions, and public key infrastructure is used to enable the recipient to decrypt the file in the owner's Pod. Currently, the data access permissions in the permission menu are limited to 'read', 'write' and 'control', however the access permissions can be extended to express access permissions to protect their privacy, intellectual property rights, and organisational data governance arrangements, including First Nations data governance or data sovereignty principles.

## 4. Conclusion

This paper describes the implementation of a Trust No One data management environment for security and privacy using encryption and transparent easy to use user interface for sharing access permissions and managing access permission on sensitive data. This work demonstrates that a TNO environment can be implemented on a highly distributed data architecture of personal datastores using AES encryption to protect data at rest and public-key cryptography to enable secure data sharing between different data stores. This ensures that data stored in Pods can only be decrypted at the user's end and is only accessible with with explicit consent of the file owner/creator or delegated controller. Using decentralised personal online datastores secured with encryption and robust consent management it is possible to build digital apps and integrated data systems that return data sovereignty and empower citizens by enabling distributed control and strong governance of linked data. Future work will focus on co-design and testing expressive access permissions with restrictions to meet common use cases for protection of personal, creative, cultural data, and trialling this system in different use case scenarios, eg. to provide both data sovereignty and dynamic consent for participants in medical research clinical trials.

## Acknowledgments

The authors acknowledge the support from the following sources which contributed to the R&D presented in this work:

- Gurriny Yealamucka Health Service in Yarrabah (North Queensland) with funding from the Australian Government Department of Health's Indigenous Australians Health Program Emerging Priorities Grant.
- Australian Government's Medical Research Future Fund National Critical Infrastructure Initiative Grant for the 'Closed Loop Non-Invasive Brain Stimulation Treatment for Depression' project.

## Declaration on Generative AI

The authors have not employed any Generative AI tools.

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
