# OpenReview forum: "Implementing Encrypted Private Data Sharing between Personal Data Vaults"
_SolidProject.org/SoSy/2026/Privacy_Session — SoSy2026-Privacy Paper_

### Official Review · ~Patrick_Hochstenbach1 · 2026-02-24
**Implementing a Trust No One Security Model in Personal Data Vaults**

**Rating:** 6
**Confidence:** 5

**Review:**

This paper describes the “Trust No One” (TNO) security model for personal data vaults as implemented by Solid‑based Pods. The implemented TNO model follows Microsoft’s three Zero Trust principles: explicit verification, least‑privilege access, and assuming breaches. Least‑privilege access is achieved by encrypting data using AES with random session keys. Data is encrypted and decrypted on‑device when the application is used. Data sharing is also supported by distributing public keys.

The core of the paper describes a demonstrator app that can create textual notes, encrypt them on‑device, and store them in a Solid Pod. Within the app, the user can grant read, write, or control permissions to other parties. As defined by the Solid permission model, a party with control rights can, in turn, grant read, write, or control permissions to others. The app provides a log of the permission history that is visible for the ‘control’ party and the original owner. The original owner can revoke ‘read/write/control’ rights for any party if desired.

The text of the paper is clearly written, although one sentence appears to have been mangled:

“This provides the file owner and they delegate  to have ’control’ an auditable record of permissions, that they can query in different ways, such as people with access in a particular date time range or with particular access permissions, or searching for all granting and revoking records to a particular recipient person” (I have a hard time to parse this).

The quality of this paper is in providing a demonstrator application for the zero-trust environment with on device encryption, logging and revoking the rights.

It is not clear in a Zero-Trust environment why a file owner would like to share “their file to another person who releases that file privately to selected other people”. I assume the original file owner should need to consent to this type of sharing of data. There is little said in the paper about consent other than providing read/write/control access (missing the purpose for read/write/control access).

In a Zero Trust environment, the required trust in the app itself is not discussed. When using the application, the app asks me to grant full control over my Pod. This is far more access than I would expect in a system based on least‑privileged access. It means I must place complete trust in the app’s developer. I accept this as a limitation of how Solid apps can currently be built. However, this is an issue that needs to be discussed and raises important questions to bring to a Solid workshop. This level of control, I assume, would be in a Zero-Trust environment a serious security and legal breach.

I like the logging component of the application, as it provides explicit verification of some of the steps. However, I miss a discussion in the paper about how the assumption of breaches has been addressed.

The paper does not provide any discussion points for the workshop or raises questions for the Solid community how to better handle Zero Trust in Solid environments. For instance, in a decentralized setting one would assume it should be possible to use some specification to allow different implementations to handle encrypted content, logs. I would like to see discussion points raised for the workshop in the paper.

---

### Official Review · ~Rui_Zhao15 · 2026-03-05
**A sensible approach for encrypted data storage and targeted sharing on Solid**

**Rating:** 7
**Confidence:** 3

**Review:**

This paper explores an interesting and important idea of storing (and sharing) encrypted data in/from you Solid Pod -- usual practices of Solid Pods trusts the Pod providers (to store clear-text or sensitive data), while this one challenges that. It contains some details in terms of how the encryption and encrypted data sharing is conducted, and claimed that there is no storage of master keys in Pods. It also briefly mentioned the idea of performing advanced (usage / access control) policy over the encrypted and shared data, and its potential advantages (without implementations, etc, though).
I am willing to support its inclusion to the session, as the corresponding flutter package does exist and the parts about encryption seem to work.

Since it's an interesting paper that I want to understand, I would also like to highlight some potential pitfalls of the paper, especially about Figure 1, which appeared to be quite informative but I find hard to interpret:

1. Despite claiming no master key is stored in Solid Pod (see top of Page 2), Figure 1 indicates otherwise ("[Bob's Pod] [Has] [Master Key]"). It should be clarified.
2. Does the "lock" icon indicate the resource/element is encrypted?
3. Why is there a dangling lock icon under "ACL"? Are ACLs encrypted, or are the file paths encrypted?
4. There are some white "keys" for "file path" and "ACL". What are those keys? The public key of Bob (to whom the data is shared)?
5. I presume the "data file", "session key" and "master key" things indicate how the data encryption is performed for any data stored in any Pods (for Bob and others). To me, it feels quite confusing that they are explained as Bob's content, while data is shared with Bob  (so they are not used in the rest of the figure).
6. Why is ACL encrypted? I mean, who performed the access control check? Bob? Why can we *trust* Bob to obey the ACLs (while the paper claimed to "trust no one")?
7. What do the Session Keys do? I mean, how can the other people share a Session Key, which is assumed to be random (at least for Bob)?
8. Are the Master Key and Session Keys for symmetric encryption or asymmetric encryption?
9. (Not about Figure 1) The paper mentioned consent and related concepts, and described some existing work (e.g. [10] and [11]), but it's not clear how they are related to the approach described in this paper -- are they complementary, predated, more advanced, or parallel?

(p.s. My rating is based on the assumption that there is no intentional factual errors, and those mentioned above can be addressed easily, given the existence of the library.)

---

### Decision · Program_Chairs · 2026-03-09

Accept (Paper)